# Consensus-Driven Evaluation of Current Practices and Innovation Feasibility in Chronic Brain Injury Rehabilitation

**DOI:** 10.3390/healthcare13212725

**Published:** 2025-10-28

**Authors:** Helena Bascuñana-Ambrós, Lourdes Gil-Fraguas, Carolina De Miguel-Benadiba, Jan Ferrer-Picó, Michelle Catta-Preta, Alex Trejo-Omeñaca, Josep Maria Monguet-Fierro

**Affiliations:** 1Physical Medicine and Rehabilitation Department, Sant Pau University Hospital, Campus Salut Barcelona, 08041 Barcelona, Spain; 2Spanish Society of Physical Medicine and Rehabilitation (SERMEF), 28016 Madrid, Spain; lgilf@sescam.jccm.es (L.G.-F.); carolina.demiguel.benadiba@sermef.es (C.D.M.-B.); 3Physical Medicine and Rehabilitation Department, Guadalajara University Hospital, 19002 Guadalajara, Spain; 4Physical Medicine and Rehabilitation Department, Ramón y Cajal University Hospital, 28034 Madrid, Spain; 5Innex Labs, Institut pel Futur, 08800 Vilanova i la Geltrú, Spain; jan.ferrer.pico@upc.edu (J.F.-P.); michelle@innex.io (M.C.-P.); alex@innex.io (A.T.-O.); jm.monguet@upc.edu (J.M.M.-F.); 6Departament d’Enginyeria Gràfica i Disseny, Escola Tècnica Superior d’Enginyeria Industrial de Barcelona (ETSEIB), Universitat Politècnica de Catalunya, 08028 Barcelona, Spain; 7Departament d’Enginyeria Gràfica i Disseny, Escola Politècnica Superior d’Enginyeria de Vilanova i la Geltrú (EPSEVG), Universitat Politècnica de Catalunya, 08800 Vilanova i la Geltrú, Spain

**Keywords:** chronic brain injury, rehabilitation, patient-centered care, tele-rehabilitation, eDelphi, consensus, Spain

## Abstract

Background: Chronic Brain Injury (CBI) is a lifelong condition requiring continuous adaptation by patients, families, and healthcare professionals. Transitioning rehabilitation toward patient-centered and self-management approaches is essential, yet remains limited in Spain. Methods: We conducted a two-phase consensus study in collaboration with the Spanish Society of Physical Medicine and Rehabilitation (SERMEF) and the Spanish Federation of Brain Injury (FEDACE). In Phase 1, surveys were distributed to patients (214 invited; 95 complete responses, 44.4%) and physiatrists (256 invited; 106 valid responses, 41.4%) to capture perceptions of current rehabilitation practices, including tele-rehabilitation. Differences and convergences between groups were analyzed using a Synthetic Factor (F). In Phase 2, a panel of 21 experts applied a real-time eDelphi process (SmartDelphi) to assess the feasibility of proposed innovations, rated on a six-point Likert scale. Results: Patients and professionals showed both alignment and divergence in their views. Patients reported lower involvement of rehabilitation teams and expressed more reluctance toward replacing in-person care with telemedicine. However, both groups endorsed hybrid models and emphasized the importance of improved communication tools. Expert consensus prioritized feasible interventions such as online orthopedic renewal services, hybrid care models, and educational video resources, while less feasible options included informal communication platforms (e.g., WhatsApp) and bidirectional teleconsultations. Recommendations were consolidated into five domains: (R1) systemic involvement of rehabilitation teams in chronic care, (R2) patient and caregiver education, (R3) self-management support, (R4) communication tools, and (R5) socialization strategies. Conclusions: This study demonstrates the value of combining patient and professional perspectives through digital Delphi methods to co-design innovation strategies in CBI rehabilitation. Findings highlight the need to strengthen communication, provide structured education, and implement hybrid care models to advance patient-centered rehabilitation. The methodology itself fostered engagement and consensus, underscoring its potential as a tool for participatory healthcare planning.

## 1. Introduction

### 1.1. Chronic Brain Injuries and the Transition Toward Self-Management

Chronic Brain Injury (CBI) can be defined as a long-term health condition resulting from acquired brain damage—including stroke, traumatic brain injury (TBI), hypoxic events, or intracranial tumors—that persists beyond the acute and subacute recovery phases. Rather than representing a single event, CBI constitutes a chronic disorder characterized by sustained neurological, cognitive, behavioral, and functional impairments that require ongoing rehabilitation and long-term management. Patients may develop late complications or experience progressive decline, highlighting the need for comprehensive and integrated models of care [1,2,3].

Individuals with CBI often experience a progressive decline in quality of life, which necessitates a proactive and dynamic rehabilitation approach. They should therefore be regarded as chronic patients, with rehabilitation strategies designed to support lifelong interaction with health and rehabilitation services. Framing them in this way underscores their progressive nature and the need for ongoing adjustments in care [4].

Given its chronic nature, CBI requires fundamentally different approaches compared with acute care. The main objectives shift toward promoting long-term well-being and quality of life, emphasizing empowerment and self-management. Evidence demonstrates that these strategies improve patient and family satisfaction, clinical outcomes, adherence, and cost-effectiveness [5,6]. Systematic reviews confirm that person-centered interventions in serious physical illness enhance both outcomes and efficiency, reinforcing the rationale for adopting self-management-oriented models in CBI rehabilitation [6].

Transitioning rehabilitation practice toward self-management models requires rethinking professional–patient relationships, dynamics, and roles. Patients must first develop health literacy regarding their condition to support active participation in care decisions. Effective communication and shared decision-making are essential to improving satisfaction and adherence. Studies have shown that strengthening patient engagement and empowerment significantly enhances perceived care quality [7,8]. However, achieving meaningful shared decision-making demands structural change that systematically incorporates patient preferences, as recommended by recent frameworks for chronic care [8].

To successfully implement these strategies, healthcare systems must explicitly integrate the lived experiences of both patients and professionals. Iterative improvements grounded in patient feedback pave the way for genuinely patient-centered care [9]. However, major shortcomings persist in systematically collecting and using patient experience data—particularly in real time—to guide rehabilitation improvement. This gap is especially pronounced in CBI, where patients face prolonged and complex care pathways.

From an epidemiological perspective, the Spanish prevalence of CBI—estimated at approximately 0.88% of the population—is consistent with figures reported in other European contexts [10]. Stroke remains the leading cause of acquired brain injury, and despite advances in acute management, a substantial proportion of survivors live with chronic neurological sequelae that require long-term rehabilitation [3]. In parallel, traumatic brain injury continues to impose a high burden, with incidence rates ranging between 47 and 849 per 100,000 population annually across European countries [11]. In the United Kingdom, more than 1.5 million people live with long-term consequences of TBI, while in Germany, approximately 300,000 new cases of acquired brain injury are reported annually, many of which evolve into chronic disability [11]. These data emphasize that CBI is not only a Spanish concern but also a major European public health challenge, reinforcing the international relevance of developing innovative and sustainable models of long-term rehabilitation.

Equally important, caregivers play a pivotal role in sustaining long-term rehabilitation for individuals with CBI. They frequently assume responsibilities for daily support, therapeutic follow-up, and coordination with health professionals. Evidence demonstrates that caregiver involvement improves functional outcomes and patient satisfaction but also generates significant physical, psychological, and financial burden [3,8]. Consequently, any patient-centered rehabilitation strategy must explicitly integrate caregiver education and support, recognizing them as indispensable partners in chronic brain injury care.

### 1.2. Spanish Context of Experience Collection in CBI Care

Over recent decades, healthcare systems worldwide have progressively shifted from paternalistic models toward patient-centered approaches that emphasize autonomy and empowerment [12]. In Spain, however, this transition has been slower, particularly in rehabilitation. As highlighted in the most recent Health System Review for Spain, reforms supporting patient involvement and integrated care remain limited and unevenly implemented [13,14]. Meaningful efforts to involve patients and professionals in shaping rehabilitation policy have only recently emerged [15]. While recognition of patient autonomy has grown, major professional organizations—including those in physical and rehabilitation medicine—acknowledge that patient perspectives are still insufficiently integrated into rehabilitation planning. This limitation is particularly evident in CBI, where platforms for structured patient–professional communication remain underdeveloped.

Nevertheless, even sporadic initiatives to gather and act upon patient and professional experiences can generate valuable lessons for advancing safety, quality, and satisfaction [16]. Such initiatives are especially relevant considering that approximately 0.88% of the Spanish population lives with CBI [10]. While several programs in Spain and abroad have promoted patient-centered care, none have explicitly targeted long-term rehabilitation in CBI.

To help address these gaps, the present study aimed to stimulate dialogue between patients and professionals, systematically capturing their experiences and jointly identifying priorities for long-term improvements in CBI rehabilitation services. In this conduct, we sought to highlight shared concerns, uncover divergences, and generate actionable recommendations for innovation.

## 2. Objectives

The primary aim of this study is to achieve consensus and establish priorities for improving care for individuals with Chronic Brain Injury (CBI) by integrating the perspectives of patients, professionals, and experts. To accomplish this, the study pursues two specific objectives:To identify experiences and needs in CBI rehabilitation by simultaneously collecting the views of patients and professionals. This includes comparing areas of agreement and divergence in their satisfaction with current models of care, as well as assessing perceptions of emerging technologies and communication channels.To propose and evaluate strategies for service improvement that promote the transition toward hybrid models (in-person and remote). This involves developing approaches for managing educational and communication resources and assessing the feasibility of these recommendations through a structured expert consensus process. The ultimate goal is to establish realistic and actionable priorities for clinical practice.

## 3. Materials and Methods

### 3.1. Study Design and Context

This study was led by an interdisciplinary team composed of three senior physiatrists, each with more than 25 years of experience in CBI rehabilitation, together with two engineers and two designers specialized in digital research platforms and participatory methodologies.

The project was carried out in close collaboration with the Spanish Society of Physical Medicine and Rehabilitation (SERMEF)—the leading professional organization for rehabilitation specialists in Spain—and the Spanish Federation of Brain Injury (FEDACE), a national network representing patients and families. SERMEF plays a key role in shaping rehabilitation policy and practice, while FEDACE serves as the primary advocacy organization for people with CBI and their caregivers.

The methodological framework followed a two-phase design (Figure 1):Phase 1: Assessment of current rehabilitation experiences through surveys administered to both patients and professionals.Phase 2: Consensus-building on the main challenges and potential improvements, enabling the prioritization of feasible recommendations for optimizing CBI rehabilitation services.

In both phases, the process began with an initial list of key rehabilitation experiences proposed by the Research Team (RT).

### 3.2. Phase 1: Assessment of Current Rehabilitation Experience

#### 3.2.1. Identification of Key Rehabilitation Experiences

The first stage of the study focused on identifying the most relevant experiences in CBI rehabilitation, ensuring that the research addressed the challenges faced by both patients and professionals. An initial list of rehabilitation experiences was compiled, covering both existing challenges in rehabilitation and emerging opportunities, particularly in tele-rehabilitation. This process is summarized in Figure 1.

This preliminary list was reviewed and refined through a focus group composed of 15 physiatrists specializing in CBI and 4 experts from FEDACE. Following this review, a final set of 24 key experiences was defined and categorized as follows:11 experiences related to current rehabilitation practices.13 experiences focused on new models of tele-rehabilitation.

#### 3.2.2. Survey Development and Data Collection

To capture the perspectives of both patients and professionals, two parallel surveys were designed and implemented:Patient Survey: Distributed to individuals with a CBI diagnosis of at least six months post-injury. Recruitment was facilitated through participating physiatrists and the FEDACE national patient network. In Phase 1, a total of 214 patients were invited, of whom 95 completed the survey in full (response rate 44.4%).Professional Survey: Sent to 256 physiatrists affiliated with SERMEF, all actively engaged in CBI rehabilitation. Screening questions ensured that only specialists with relevant expertise were included in the final analysis. In Phase 1, 106 valid responses were obtained (response rate 41.4%). These surveys gathered perceptions of current rehabilitation practices, including the use of tele-rehabilitation.

Both surveys employed closed-ended questions rated on a six-point Likert scale, enabling participants to provide quantifiable assessments of rehabilitation experiences and service challenges. All items were scored using differential semantics, where 1 indicated “very little” and 6 indicated “very much.” For example, for the question “To what extent does the rehabilitation team participate in hospital discharge care?”, a score of 1 reflected minimal participation, whereas a score of 6 indicated extensive involvement.

Additionally, demographic data were collected, including age, gender, geographic location, professional experience (for physiatrists), and time since injury (for patients). The surveys remained open for four months.

#### 3.2.3. Inclusion and Exclusion Criteria

For the patient group, inclusion criteria were: (1) diagnosis of chronic brain injury (CBI) of any etiology; (2) at least six months elapsed since diagnosis; (3) age ≥ 18 years; and (4) ability to complete an online survey independently or with caregiver support. Exclusion criteria included: (1) acute or subacute stages of brain injury (<6 months from diagnosis); (2) absence of informed consent through participation; and (3) incomplete survey submission.

For the professional group, inclusion criteria were: (1) being a specialist in Physical Medicine and Rehabilitation (physiatrist) and (2) currently providing care to CBI patients within Spain. Exclusion criteria were: (1) professionals from non-rehabilitation disciplines or (2) incomplete survey data.

#### 3.2.4. Descriptive and Concordance Analysis

Once the survey data were collected, the responses underwent a visual–descriptive assessment and triangulation process conducted by the research team (Figure 2). This analytical stage focused on identifying discrepancies between patients and professionals in their perceptions of challenges and priorities, while also highlighting areas of agreement that could serve as a foundation for the consensus-building phase in Phase 2. Using frequency distributions and comparative charts, the research team translated the survey findings into actionable insights to guide the next stage of the study.

Descriptive analyses included frequency distributions and percentages of agreement, defined as the proportion of participants rating an item ≥ 4 on the six-point Likert scale. Concordance between patients and professionals was quantified using the Synthetic Factor (F), a composite measure that integrates both mean differences and agreement rates. This approach was designed to highlight the items with the greatest divergence, ensuring that discrepancies were captured both in absolute Likert scores and in agreement proportions.

The survey instruments were validated prior to distribution. Specifically, a panel of 17 expert physiatrists from SERMEF and 4 patient representatives from FEDACE reviewed all items for clarity, clinical relevance, and feasibility. This validation ensured content validity and appropriateness for both target groups. While no separate pilot survey was conducted, the iterative expert–patient validation process served to refine the items before full deployment. To enhance transparency, a selection of survey items (in their original wording for both patients and professionals) is provided as Appendix A. These examples illustrate how perceptions of care and rehabilitation experiences were operationalized in the questionnaires.

### 3.3. Phase 2: Evaluation of Innovation Feasibility

#### 3.3.1. Consensus-Building via e-Delphi

The second phase of the study aimed to establish expert consensus on the feasibility of implementing specific improvements in CBI rehabilitation.

A second focus group was convened, consisting of 21 physiatrists specializing in CBI, who reviewed and prioritized the key areas identified in Phase 1. From this process, 14 innovation priorities in CBI management were defined.

To structure decision-making, we employed a real-time digital adaptation of the Delphi method, the 2025 version of the SmartDelphi platform [17]. This tool enabled participants to receive immediate aggregated feedback after submitting their responses, allowing them to revise their answers and, if desired, modify their votes. This approach reduced dropout rates and provided a detailed record of how opinions shifted within a single round [18,19]. The use of real-time Delphi panels offers several practical advantages, including stability, rapid consensus, and continuous feedback [20].

Voting and discussion were conducted during a synchronous, in-person working session. This method allowed participants to:Receive immediate and aggregated feedback, encouraging iterative reflection and adjustment of responses.Engage in multiple “mini-iterations” within a single Delphi round, increasing the accuracy and reliability of the consensus outcomes.

Experts were recruited through SERMEF, ensuring broad national representation across several autonomous communities (including Catalonia, Madrid, Andalusia, Valencia, and Castilla-La Mancha). The gender distribution was balanced (11 women and 10 men), and all participants had substantial clinical experience (mean 17 years, range 8–28).

#### 3.3.2. Implementation Feasibility Ratings

During the e-Delphi session, participants were asked to evaluate the feasibility of implementing proposed rehabilitation improvements using a six-point Likert scale, where 1 = very difficult to implement and 6 = highly feasible.

This phase provided a structured, data-driven foundation for the final set of recommendations. The feasibility ratings enabled the identification of areas where implementation would be relatively straightforward, as well as those requiring substantial structural changes. Figure 3 illustrates the SmartDelphi interface used to facilitate the consensus process.

## 4. Results

This study enabled the identification of significant differences in the perceptions and experiences of chronic brain injury rehabilitation between patients and professionals, while also establishing consensus on recommendations to improve these services. Across two consensus phases, perceptions of care quality were systematically analyzed, and recommendations were prioritized according to their feasibility of implementation. These findings culminated in a final proposal organized into specific domains of innovation aimed at strengthening the management of patients with chronic brain injury.

### 4.1. First Phase: Comparison of Experiences Between Patients and Professionals

The results of the first phase reveal disparities in the perception of received care and the usefulness of telemedicine tools. Table 1 presents the results of the quantitative study conducted. The perception of agreement is indicated by the percentage of patients or professionals who agree or strongly agree (scoring 4–6 on a Likert scale) with the given statement.

To enable comparison between patient and professional responses, we propose a descriptive concordance analysis based on two complementary indicators.

MP (Mean Perception) measures the distance of each item’s mean from the maximum positive agreement on the 6-point Likert scale. It reflects the joint perception of professionals and patients for each item and is calculated as:MP_i_ = 6 − (μ_i Prof._ + μ_i Pat._)/2
where μ_i_ is the mean score for the item. “A higher MP value indicates that, on average, both groups are farther from the maximum possible agreement (6), i.e., the statement is perceived more negatively. Note that a higher MP does not mean more positive perception, but rather a larger distance from the ideal maximum (6).

MD (Mean Difference) captures the discrepancies between patients and professionals in terms of both central tendency and agreement. It is defined as:MD_i_ = (μ_i Prof._ − μ_i Pat._) × (A_i Prof._ − A_i Pat._)
where A_i_ is the percentage of patients or professionals who agree or strongly agree (scoring 4–6 on a Likert scale). We chose the product of the two differences to ensure that both a divergence in mean scores and a divergence in high-agreement rates are necessary to produce a large MD. If one of them is small, the product remains moderate, which prevents spurious discrepancies. On the other hand, this strategy was selected against other alternative formulations because it facilitates interpretation and allows expert participants in the next phase to easily grasp the meaning of F.

Combining MP and MD, we introduced the Synthetic Factor (F), a composite metric that is the maximum of two values that simultaneously accounts for the joint distance of responses and the discrepancies between patients and professionals.F_i_ = Max (MP_i_; MP_i_ × MD_i_/10)

This definition ensures that we consider the worst case in the assessment of each statement. The scalar 10 was chosen after testing different normalizations to keep values within the same range as the original Likert scale and to prevent overinflated results. The use of the maximum operator was preferred over averaging to increase sensitivity: significant issues are highlighted, whether they stem from low joint perception or from strong discrepancies.

A simplified example can facilitate the interpretation of the synthetic factor (F): A recommendation such as “providing home exercise videos” is rated by experts with values ranging from 4 to 6; the synthetic factor (F) would be close to the maximum (near 6). This indicates that, in practical terms, the group perceives this recommendation as very easy to implement. Conversely, if ratings are spread between 2 and 5, the resulting F would be lower, reflecting that experts foresee more difficulties in real-world implementation. In this way, F serves as a simple summary indicator of consensus feasibility, as it captures both the overall positivity of ratings and the divergence between experts’ views.

Table 1 reports, for each rehabilitation experience, the mean scores, percentage agreement, and the calculated Synthetic Factor (F).

Patients tend to rate the involvement of rehabilitation staff lower than professionals do. Likewise, there is less acceptance of telemedicine among patients, particularly regarding the replacement of in-person visits with teleconsultations. However, both patients and professionals agree on the need to improve communication and education tools to optimize at-home rehabilitation follow-up.

### 4.2. Second Phase: Validation and Prioritization of Recommendations

In the second phase, a panel of 21 experts evaluated the feasibility of implementing the proposed recommendations using a real-time Delphi process. Table 2 summarizes the recommendations and their feasibility ratings on a 1–6 scale, ordered from most to least feasible according to the statistical parameters (μ, σ, Med, IQR). N indicates the original order in the eDelphi survey, while F represents a synthetic factor derived from the comparison of the two preceding surveys, calculated as the average of the F values of the key experiences (sN) associated with each evaluated item (eN).

The most feasible recommendations included the implementation of an online orthopedic renewal service, the development of hybrid care models, and the use of reference videos for home-based rehabilitation. In contrast, the integration of informal communication platforms (e.g., WhatsApp) and the deployment of bidirectional telemedicine were considered less feasible.

In Figure 4, perception items (s1–s24) from Table 1 have been ordered by Synthetic Factor (F), which reflects the divergence between patients and professionals. Bar colors represent the mean perceived difficulty (μ) calculated from the corresponding recommendations in Table 2. This combined view highlights which items are both most divergent and most difficult to implement.

### 4.3. Final Proposal: Innovation Strategies in the Management of Chronic Brain Injury

Building on the previous results, the recommendations were consolidated into specific innovation domains. Table 3 illustrates the aggregation of these domains into coherent action packages aimed at supporting the development of targeted projects in Innovation for Chronic Brain Injury (CBI) management. The relative difficulty of implementation, expressed through the F factor, ranged from 1.3 to 3.9. Normalization was performed using min–max scaling to map the observed range into a 1–10 scale.

#### 4.3.1. R1. Strategies and Methodologies for the Systemic Involvement of the Rehabilitation Team in Chronic Patient Care

Patients currently report satisfactory involvement of the rehabilitation team at the time of hospital discharge; however, notable opportunities for improvement remain. Strengthening and formalizing this process could facilitate a more structured transition toward patient self-management, thereby enhancing adherence to therapeutic recommendations at home and ensuring continuity of follow-up. Furthermore, the development of standardized support tools for this transition would optimize communication with the care team and reduce patients’ perception of fragmented care.

#### 4.3.2. R2. Strategies and Tools for Patient and Caregiver Education

Education of patients and caregivers is a key determinant in enhancing the effectiveness of the rehabilitation process and fostering self-management. The implementation of structured guides and audiovisual resources can improve understanding of the care pathway and reduce reliance on professional support for recurrent questions. Moreover, access to demonstration videos may strengthen patients’ confidence in performing exercises correctly, thereby promoting adherence to treatment and contributing to more efficient recovery [21].

#### 4.3.3. R3. Promotion and Facilitation of Self-Management of the Rehabilitation Process by Patients and Caregivers

Self-management constitutes a fundamental pillar in the rehabilitation of individuals with chronic brain injury [5,6,9]. Combining digital and in-person tools offers greater flexibility for both follow-up and therapy delivery, reducing unnecessary travel and supporting treatment continuity [22]. In addition, the establishment of structured systems for orthopedic renewal and home adaptations can help preserve functional autonomy, minimize risks, and ultimately improve long-term quality of life [23].

#### 4.3.4. R4. Strategies and Tools to Improve Communication with Patients and Caregivers

One of the main challenges in the management of chronic brain injury is ensuring agile and effective communication between professionals, patients, and caregivers [5,6,9]. The use of digital channels, such as WhatsApp or dedicated applications, may facilitate more immediate responses to recurring queries, thereby improving the perception of support and reducing the need for in-person visits [24]. In parallel, the integration of bidirectional video consultations could increase the efficiency of therapeutic supervision and foster greater patient engagement in the rehabilitation process.

However, the use of informal messaging tools such as WhatsApp also entails relevant risks. These include privacy and data-protection issues, lack of integration into official electronic health records, medico-legal uncertainty related to traceability and informed consent, and potential inequities in digital access. Therefore, while these channels may improve immediacy and patient engagement, their adoption should occur within institutional frameworks that ensure security, proper documentation, and equity of use.

#### 4.3.5. R5. Promotion and Facilitation of Socialization Strategies Among Patients and Caregivers

The social dimension represents a crucial component of well-being in individuals with chronic brain injury [5,9,10]. Establishing both in-person and virtual spaces for sharing experiences and challenges with peers can help reduce feelings of isolation and enhance motivation throughout rehabilitation. Furthermore, promoting socialization among patients and caregivers enables the exchange of best practices and the strengthening of support networks, thereby facilitating more effective adaptation to the long-term realities of chronic conditions.

## 5. Discussion

This study provides new insights into the current state of CBI rehabilitation in Spain by systematically capturing and comparing the perspectives of patients and professionals, and by reaching expert consensus on feasible innovations for service improvement. The results highlight both convergence and divergence in views, underscoring the complexity of designing rehabilitation strategies that are simultaneously patient-centered, clinically effective, and systemically feasible.

A key finding concerns the limited acceptance of telemedicine among patients, particularly when remote modalities are proposed as substitutes for in-person visits. This reluctance reflects the well-documented value of face-to-face encounters for assessments, adjustments, and hands-on therapeutic activities [24,25]. However, both patients and professionals expressed support for hybrid models, echoing international trends that emphasize the integration of digital health tools with traditional care pathways [25]. Evidence from randomized controlled trials, such as the VIGoROUS study [21], has already demonstrated the clinical potential of self-managed, technology-supported rehabilitation. Nonetheless, patient perceptions in our study suggest that careful attention must be paid to tailoring digital solutions to individual needs and capacities, thereby avoiding the perception that technology diminishes rather than enhances care quality.

Another major theme is the critical role of communication. Our findings reveal gaps in how patients perceive the involvement of rehabilitation teams, particularly during transitional moments such as hospital discharge. While professionals reported strong engagement, patients perceived this involvement as weaker. This discrepancy has also been noted in prior studies, where effective patient–professional communication and shared decision-making were found to be essential for patient satisfaction and adherence [7]. Bridging this gap will require not only interpersonal skills training but also structural reforms that embed systematic patient feedback into service evaluation [9].

Patient education emerged as another area of divergence. Although prior research consistently demonstrates that structured educational strategies improve engagement, adherence, and outcomes [26], a substantial proportion of both patients and professionals in our study did not view educational video content as essential. This finding raises questions about cultural and systemic barriers to patient empowerment in Spain, where rehabilitation services have traditionally been provider-driven [8,9]. Future initiatives should explicitly address these barriers by developing tailored educational resources that align with patients’ health literacy levels and preferences.

Our results also resonate with broader European debates on chronic care reform. As highlighted by Legido-Quigley et al. [12] and Bernal-Delgado et al. [13], Spain has been slower than other health systems in implementing integrated, people-centered approaches. The limited incorporation of patient perspectives in rehabilitation policy, acknowledged by national reports [14,15], may partly explain the gaps identified here. By contrast, systematic reviews demonstrate that person-centered interventions yield improvements in both patient outcomes and cost-effectiveness [16]. Therefore, embedding participatory methods—such as the digital Delphi approach used in this study—represents a promising pathway for aligning rehabilitation services with international best practices.

Finally, the methodological contribution of this study deserves emphasis. By employing a real-time eDelphi process [18,19,20], we were able to minimize participant attrition, generate immediate feedback loops, and foster consensus within a single round. Participants reported that being consulted in this structured way was itself empowering, reinforcing earlier findings that inclusive methodologies can function as interventions in their own right, promoting engagement and trust [19].

In summary, our study demonstrates that meaningful progress in CBI rehabilitation in Spain will require a combination of strategies:strengthening communication and shared decision-making [7,8],integrating hybrid rehabilitation models that balance in-person and remote care [22],expanding structured patient education initiatives [5,6,26], andembedding systematic patient and professional feedback into service design [9,16].

Together, these steps would advance the shift toward genuinely patient-centered, sustainable models of CBI rehabilitation.

## 6. Study Limitations

This study has several limitations that should be acknowledged when interpreting its findings. First, although participants were recruited from across Spain, the results may not be fully generalizable to other countries or healthcare systems with different organizational structures and policy frameworks. Comparative research across diverse health systems would be needed to confirm the transferability of our conclusions [12,13].

Second, the survey sample may reflect a degree of self-selection bias. Patients were recruited through advocacy networks (FEDACE) and professionals through SERMEF, which could have favored individuals more engaged with rehabilitation services. While we attempted to validate representativeness by comparing demographic profiles with published epidemiological data, gender imbalances and potential underrepresentation of certain subgroups (e.g., rural patients, non-affiliated professionals) remain possible [10].

Third, the study relied primarily on quantitative survey data, which limited the ability to capture the depth and nuance of personal experiences. Although we mitigated this by consulting experts and patient organizations during the preparatory phase, future studies would benefit from incorporating qualitative methods (e.g., in-depth interviews or focus groups) to provide richer insights into lived experiences [9].

Fourth, while the study generated prioritized recommendations, it did not address the operational details of implementation. The feasibility ratings obtained in the eDelphi process reflected the perspectives of rehabilitation professionals but did not fully consider system-level barriers such as administrative constraints, funding limitations, or technological infrastructure. Including these dimensions would provide a more comprehensive picture of implementation challenges [23,25].

This was a cross-sectional study capturing perceptions at a single point in time. Given that attitudes toward rehabilitation and telemedicine are likely to evolve—especially as digital health adoption increases—longitudinal research with repeated measurements and iterative feedback loops would be essential to track changes and inform adaptive service models [21,26].

Finally, although both questionnaires were reviewed by experts and patient representatives to ensure content validity, they were newly designed for this study and not formally validated, which should be considered a limitation.

Despite these limitations, the study contributes valuable knowledge to the understanding of CBI rehabilitation in Spain. It demonstrates the feasibility and added value of participatory, consensus-driven methodologies for engaging patients and professionals, and provides a foundation for future research aimed at strengthening patient-centered rehabilitation.

## 7. Conclusions

This study highlights both areas of convergence and divergence between patients with chronic brain injury (CBI) and rehabilitation professionals in Spain, offering a valuable foundation for advancing long-term models of care. By applying a real-time eDelphi approach, we identified shortcomings in current services as well as opportunities for innovation, particularly in hybrid models that integrate in-person and remote rehabilitation while strengthening patient self-management.

The findings underscore three priority areas for action:Enhancing communication and shared decision-making between patients and professionals to improve satisfaction, trust, and adherence [6,7].Expanding structured educational resources tailored to patients and caregivers, which are critical to fostering empowerment and health literacy [5,6,26].Developing flexible, hybrid rehabilitation models that balance the strengths of face-to-face care with the scalability of digital health tools [21,23,25].

Implementing these strategies would not only address current gaps in Spanish rehabilitation services but also align national practice with international standards for patient-centered, cost-effective care [6,12,13]. Moreover, the study demonstrates the added value of participatory research methodologies, showing that the process of engaging patients and professionals in decision-making can itself promote empowerment and trust [9,18].

Looking forward, the next step will be to pilot the prioritized recommendations in real-world settings, supported by adequate resources and technical assistance. Systematic evaluation of outcomes and satisfaction, combined with iterative adjustments informed by continuous feedback, will be essential to sustain improvements over time. Expanding these participatory approaches beyond Spain could further validate their relevance and adaptability across different health system contexts.

In conclusion, this study provides evidence that combining patient and professional perspectives through digital consensus methods can accelerate the transition toward genuinely patient-centered rehabilitation in CBI. By strengthening communication, education, and hybrid care models, rehabilitation services can be reoriented to better meet the evolving needs of individuals living with chronic brain injury.

Beyond the prioritization of actions, the dual perspective provides a unique evidence base for designing rehabilitation pathways that are not only clinically sound but also socially legitimate. Such consensual recommendations may serve as a valuable input for health policy and strategic planning, helping to align rehabilitation services with both patient needs and system-level priorities.

## Figures and Tables

**Figure 1 healthcare-13-02725-f001:**
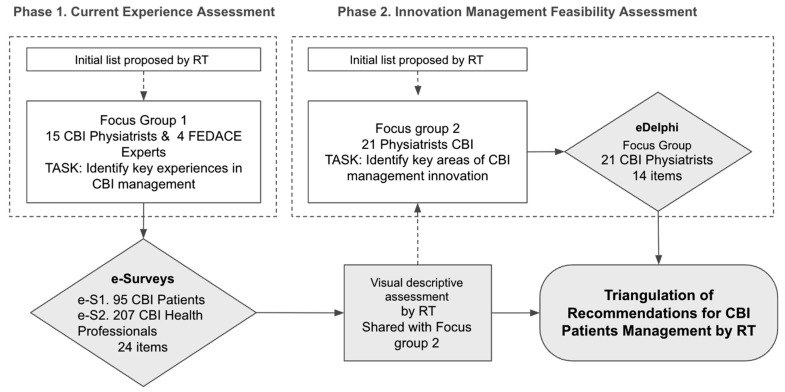
Flow of the two phases of research.

**Figure 2 healthcare-13-02725-f002:**
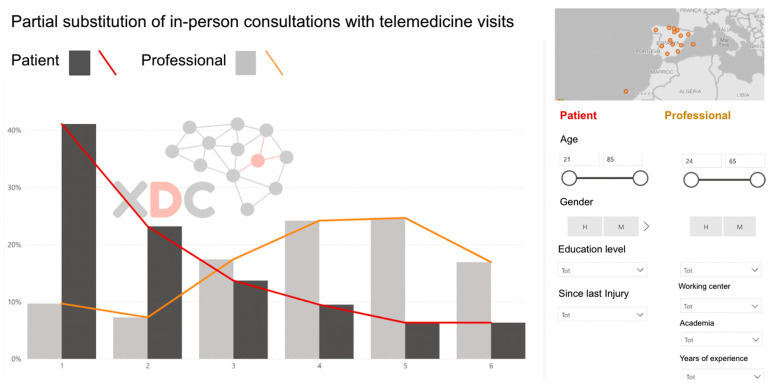
Sample screen of the 24 visual descriptive analysis to compare the surveys of patients and professionals.

**Figure 3 healthcare-13-02725-f003:**
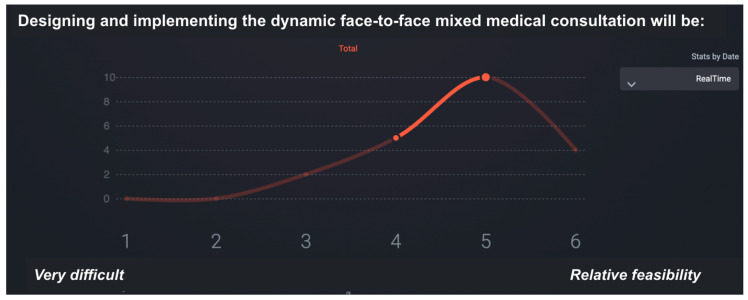
Sample screen of the SmartDelphi tool used to facilitate consensus in the eDelphi synchronous focus group. The median is 5 and IQR has a value of 1.

**Figure 4 healthcare-13-02725-f004:**
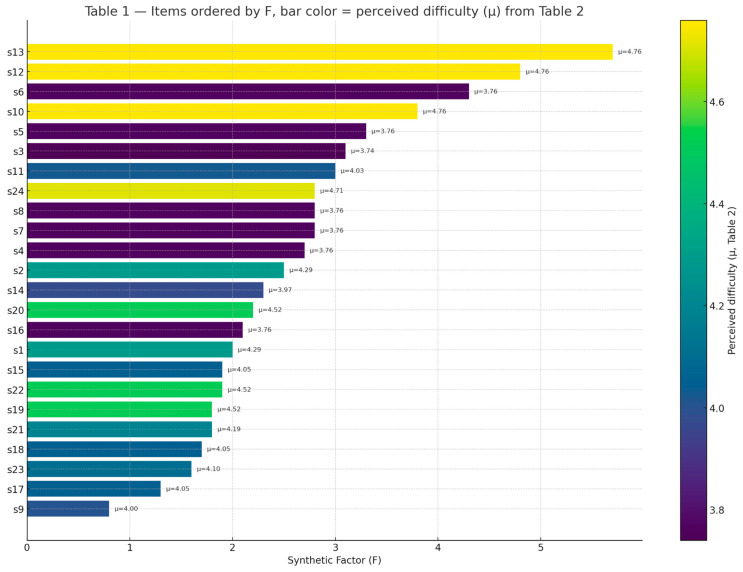
Combined representation of data from Table 1 and Table 2, reflecting the coherence between the lack of agreement between patients and professionals and the relative difficulty of innovation implementation.

**Table 1 healthcare-13-02725-t001:** Differences in perception between patients and professionals.

		Patients		Professionals		Factor
	Category	Mean	Agree	Mean	Agree	(F)
	Current experiences in CBI rehabilitation	4.4	58.5	5.1	81.9	2.2
s1	Perceived involvement of the rehab team during hospitalization.	4.2	56.8	5	78.7	2.5
s2	Perceived involvement of the rehab team during discharge.	4.5	60.1	5.2	85.1	2.0
	Current and past experiences of hybrid telemedicine and telerehabilitation	3.6	45.8	4.0	57.8	2.2
s3	Satisfaction with in-person check-ups.	4.6	72.6	4.7	74.9	1.4
s4	Satisfaction with remote consultations.	3.2	33.7	3.5	38.8	2.7
s5	Approval of remote assistance (during COVID-19).	3.8	45.3	4.2	66.7	2.0
s6	Usefulness of remote interaction with rehab services	3.5	38.9	4.1	71.1	4.3
s7	Practicality of remote interactions	3.4	37.9	4	66.7	4.0
s8	Acceptance to replace in-person visits with telemedicine	2.7	22.1	3.2	34.3	3.1
s9	Approval of home-based rehabilitation	5.1	94.5	5.3	92.3	0.8
s10	Acceptance of remote orthopedic adjustments	3	38.9	3.5	50.1	2.8
s11	Acceptance of remote home adaptations	3.2	28.4	3.6	25.6	2.6
	Communication channels for new telerehabilitation services	4.0	57.8	4.5	70.2	2.7
s12	Preference for face-to-face interactions	4	57.9	5	89.9	4.8
s13	Support for phone/video checkings	3.6	49.5	4.5	82.1	5.7
s14	Approval of online communications	4.2	71.6	4.8	80.2	1.5
s15	Preference for email as the communication channel	3.5	43.5	4	50.7	2.3
s16	Preference for WhatsApp as the communication channel	4	58.9	3.5	33.8	2.8
s17	Support for hybrid rehabilitation services	4.5	65.3	5.2	84.5	1.5
	Resources and education in telerehabilitation	3.9	62.0	4.3	79.1	1.9
s18	Need for tele-rehabilitation tutorials	4.1	67.4	4.4	70.1	1.8
s19	Lack of resources for recording/sharing rehab videos	3.8	62.1	4	72.9	2.1
s20	Support for home exercise programs	4.3	72.7	4.7	85.5	1.5
s21	Support for demonstration videos	4.2	71.6	4.6	86	1.6
s22	Support for health education videos	3.9	58.9	4.3	81.6	1.9
s23	Interest in shared experience spaces	3.7	52.6	4.2	86.5	3.5
s24	Interest in group therapy sessions	3.5	48.4	4	71.1	2.6
	Totals	3.9	54.7	4.3	69.6	1.9

s1–s24 refers to the 24 statements (survey items) included in the survey, covering current rehabilitation experiences (s1–s11) and tele-rehabilitation experiences (s12–s24).

**Table 2 healthcare-13-02725-t002:** List of areas of innovation in the management of CBI patients.

N	Question	μ	σ	Med	IQR	F	sN
e12	Formalizing and implementing the non-face-to-face service for orthopedic adjustments will be:	5.19	0.93	5	1	2.8	s10
e8	Designing and implementing the dynamics of mixed face-to-face–non-face-to-face medical consultation will be:	4.76	0.89	5	1	2.8	s9 to s14
e10	Providing model videos so that the patient can follow the therapies in a non-face-to-face mode will be:	4.67	1.11	5	1	1.8	s10 to s22
e14	Enabling education programs for patients to improve their interaction with the PM&R service will be:	4.43	0.93	5	1	1.8	s18
e7	Designing and implementing the dynamics of mixed face-to-face–non-face-to-face therapy will be:	4.33	1.06	4	2	3.9	s12 s14 s17
e1	Formalizing the involvement of the rehabilitation team in patient care upon discharge from the hospital will be:	4.29	0.9	4	1	2.0	s2
e13	Strengthening the advisory capacity of the PM&R service in relation to home adaptations will be:	4.05	0.59	4	0	1.7	s9 s11
e9	Designing the process of supporting patients in the follow-up of their therapies at home will be:	3.62	0.86	4	1	3.8	s9 s12 s13
e3	The patient/caregiver understands that the sequence of steps to follow in the care process in the rehabilitation service will be:	3.62	0.97	3	1	1.3	s9 s18
e2	Formalizing the involvement of the rehabilitation team in patient care when the patient is already chronic will be:	3.57	0.93	3	1	3.1	s3 s12
e4	Formally structuring the programs and communication for carrying out therapy exercises at home will be:	3.29	0.85	3	1	1.6	s18 s20
e6	Implementing video communication between the patient/caregiver and the professional in both directions will be:	3.24	1.18	3	2	2.1	s19
e11	Enabling patients to share among themselves their experiences related to the rehabilitation process will be:	3.14	1.2	3	2	3.0	s23 s24
e5	Adapting to the use of WhatsApp or a similar application as a communication tool with the patient/caregiver will be:	3.05	1.16	3	2	2.3	s15

e1–e14 refers to the 14 recommendations evaluated, ordered from most to least feasible.

**Table 3 healthcare-13-02725-t003:** Main recommendations for Specific Projects in Innovation in CBI Patient Management.

	Areas of Innovation	Level of Difficulty and Recommendation
R1	Strategies and methods for the systemic involvement of the rehabilitation team in chronic patient care: ● Upon hospital discharge ● During patient follow-up	5. Moderate The rehabilitation team is already involved, but strengthening the transition toward the patient’s self-management is needed.
R2	Strategies and tools for patient and caregiver education: ● Understanding the sequence of steps in their rehabilitation process ● A guide for interaction with rehabilitation services ● Video models to follow therapy exercises	2. Low Implementing structured educational materials is straightforward and highly feasible.
R3	Promotion and facilitation of self-management in rehabilitation by patients and caregivers: ● Programs for home-based therapy exercises ● Hybrid models of face-to-face and tele-therapy ● Hybrid models of face-to-face and tele-medical consultation ● Follow-up on home therapy ● Orthopedic renewal services ● Home adaptations	6 High Combining tele-rehabilitation with home-based follow-up requires infrastructure and changes in care dynamics.
R4	Communication strategies and tools for patient and caregiver interaction: ● WhatsApp or a similar application ● Bidirectional video communication	4 Moderate Digital communication channels are viable, but they must be adapted to both patient and healthcare system needs.
R5	Promotion and facilitation of socialization strategies among patients and caregivers: ● Activities and spaces for sharing rehabilitation experiences	7 High Establishing socialization spaces requires active engagement and coordination among patients, caregivers, and professionals.

## Data Availability

The data presented in this study are available on request from the corresponding author due to privacy restrictions.

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
