# Peer review of "Consensus-Driven Evaluation of Current Practices and Innovation Feasibility in Chronic Brain Injury Rehabilitation"

_healthcare, 2025, doi:10.3390/healthcare13212725_

Round 1
Reviewer 1 Report
Comments and Suggestions for Authors
Summary
The manuscript investigates perceptions of patients and professionals regarding chronic brain injury (CBI) rehabilitation in Spain and explores feasible innovations using a two-phase consensus methodology (surveys and a real-time eDelphi). The study is well conceived, methodologically solid, and provides valuable insights into the gaps between patient and professional perspectives. Its main strengths are the participatory design, the integration of both quantitative and consensus-based methods, and the focus on patient-centered rehabilitation strategies that align with international trends.
Specific Comments
Introduction
The Spanish context is well explained, but could benefit from additional comparative epidemiological data with other European countries to highlight international relevance.
Consider adding more detail on caregiver involvement early in the introduction, as they emerge as key stakeholders later in the paper.
Methods
The explanation of the Synthetic Factor (F) is very technical. A simplified example would make it more accessible.
Please clarify whether the survey instruments were validated or piloted prior to use. It would be very useful to provide examples of the survey questions (e.g., in an appendix or as supplementary material). This would increase transparency and allow readers to better understand how perceptions were measured.
In Phase 2, more details on expert selection (e.g., geographic distribution, gender, professional profile) would strengthen confidence in the representativeness of the panel.
Results
The table 1 is informative but very dense. A visual figure (e.g., heatmap or bar chart) summarizing the main divergences between patients and professionals would improve readability.
Discussion
The discussion on WhatsApp and informal tools should include a more critical perspective. Risks such as privacy concerns, lack of integration into official health records, medico-legal implications, and unequal digital access should be acknowledged.
The role of caregivers could be discussed in greater depth. They are central in CBI rehabilitation and could strongly influence acceptance of hybrid and digital models.
Limitations
Please clarify whether the survey tools were validated, as this is a limitation if not addressed.
Author Response
Comment 1:
Specific Comments
Introduction
The Spanish context is well explained, but could benefit from additional comparative epidemiological data with other European countries to highlight international relevance.
Consider adding more detail on caregiver involvement early in the introduction, as they emerge as key stakeholders later in the paper.
Response to comment 1: We have revised the Introduction section, incorporating comparative epidemiological data and providing additional details on caregiver involvement (attached as a PDF).
Comment 2.
Methods
The explanation of the Synthetic Factor (F) is very technical. A simplified example would make it more accessible.
Please clarify whether the survey instruments were validated or piloted prior to use. It would be very useful to provide examples of the survey questions (e.g., in an appendix or as supplementary material). This would increase transparency and allow readers to better understand how perceptions were measured.
In Phase 2, more details on expert selection (e.g., geographic distribution, gender, professional profile) would strengthen confidence in the representativeness of the panel.
Response to comment 2 METHOD:
|
We added a short example at the end of section 4.1 First Phase: Comparison of Experiences between Patients and Professionals. |
|
We added a comment at the end of section 3.2.3. Descriptive and Concordance Analysis, and added an Appendix at the end of the paper with specific examples. (line 220) |
|
We have added more detail on expert selection to guarantee confidence in the representativeness of the panel. We have expanded the description of the Phase 2 expert panel in Methods at the end of 3.3.1. Consensus-Building via e-Delphi |
Comment 3: Results The table 1 is informative but very dense. A visual figure (e.g., heatmap or bar chart) summarizing the main divergences between patients and professionals would improve readability.
Response to comment 3:
|
Thank you for your valuable suggestion. In addition to Table 1, we have now included a visual figure to improve readability of the main divergences between patients and professionals. Moreover, we have connected table 1 and table 2 through the new graph. This is marked as Figure 4. |
Comment 4:
Discussion
The discussion on WhatsApp and informal tools should include a more critical perspective. Risks such as privacy concerns, lack of integration into official health records, medico-legal implications, and unequal digital access should be acknowledged.
The role of caregivers could be discussed in greater depth. They are central in CBI rehabilitation and could strongly influence acceptance of hybrid and digital models.
Response to comment 4:
|
We have revised the discussion on the use of WhatsApp and other informal digital tools to include a more critical and balanced perspective. The updated text now acknowledges the main risks associated with these tools —including privacy and data protection issues, lack of integration into electronic health records, medico-legal uncertainties, and potential inequalities in digital access— while still recognizing their perceived usefulness in clinical communication and patient support. |
|
We have included a deeper comment about caregivers in the introduction. We understand that this comment is already covered. |
Comment 5:
Limitations
Please clarify whether the survey tools were validated, as this is a limitation if not addressed.
Response to comment 5:
|
Thank you for pointing this out. The questionnaires were developed specifically for this study and were not based on previously validated instruments. We have clarified this point in the text and explicitly acknowledged it as a study limitation. Both surveys underwent content review and validation by FEDACE and a panel of 17 expert physiatrists from SERMEF to ensure face and content validity; however, no formal psychometric validation was conducted. A corresponding statement has been added in the Study Limitations section of the Discussion (line 503). |

Reviewer 2 Report
Comments and Suggestions for Authors
Dear authors,
The manuscript entitled “Consensus-Driven Evaluation of Current Practices and Innovation Feasibility in Chronic Brain Injury Rehabilitation” is bringing into attention the importance of combining both patient and professional perspectives by integrating digital Delphi methods to co-design innovative strategies in CBI rehabilitation. To be able to deliver proper care notably on the long-term, it is of paramount importance to strengthen communication, offer structured education, and implement hybrid care models to ensure continuum of care and patient-oriented rehabilitation services. While the topic is interesting, there are some aspects that should be addressed:
- What were the inclusion and the exclusion criteria applied to this study? Please add this information in the manuscript.
- Regarding Figure 3 from 3.3.2. Implementation Feasibility Ratings, it appears in the Results section. If you want it in the previous section, move it accordingly. If you want it in the Results, you have to first address it there.
- Make sure all the references are cites according to the journal`s guidelines and also verify typo in the manuscript.
- I find your study very interesting and robust, therefore, I consider it suitable for publication once you address the previous small suggestions.
Good luck!
Author Response
Comment 1:
- What were the inclusion and the exclusion criteria applied to this study? Please add this information in the manuscript.
Response to comment 1:
|
We appreciate this observation. The inclusion and exclusion criteria have now been explicitly stated in the Methods section. In brief, both target groups—patients and professionals—had specific eligibility requirements. We have added the following paragraph to clarify this. (line 205) |
Comment 2: Regarding Figure 3 from 3.3.2. Implementation Feasibility Ratings, it appears in the Results section. If you want it in the previous section, move it accordingly. If you want it in the Results, you have to first address it there.
Response to comment 3:
|
Thank you for your observation, it was a mistake. We have moved the Figure 3 to the correct position. |
Comment 4: Make sure all the references are cites according to the journal`s guidelines and also verify typo in the manuscript.
Response to comment 4:
|
Thank you for your comment. We have done that |
Comment 4: I find your study very interesting and robust, therefore, I consider it suitable for publication once you address the previous small suggestions.
Response to comment 4: We greatly appreciate your positive comments and are glad you found the study interesting and robust. All suggested revisions have been carefully implemented.
Reviewer 3 Report
Comments and Suggestions for Authors
Abstract
The abstract is well written. However, it would benefit from an early definition of chronic brain injury to provide clarity for the reader.
Introduction
- Consider defining chronic brain injury at the beginning of the section.
- Lines 51–60 lack supporting references; please add citations where appropriate.
- In lines 61–62, clarify why different approaches are required, providing a brief justification would strengthen the argument.
- There are several strikethrough lines throughout the document, please remove these for clarity.
Methods
This section is well written. However, some statements lack proper referencing. For example:
- Lines 338–339
- Lines 347–348
Please ensure that all evidence-based statements are supported with appropriate citations.
Discussion
The discussion is well written and clearly presented.
Author Response
Comment 1: Abstract
The abstract is well written. However, it would benefit from an early definition of chronic brain injury to provide clarity for the reader.
Response to comment 1: Thank you for this helpful suggestion. We have clarified the concept of chronic brain injury (CBI) in the Introduction section, providing a concise definition at the beginning to enhance conceptual clarity. We made only minimal modifications to the Abstract to maintain its length and focus, while ensuring consistency with the revised introduction.
Response to comment 2: Introduction
- Consider defining chronic brain injury at the beginning of the section.
- Lines 51–60 lack supporting references; please add citations where appropriate.
- In lines 61–62, clarify why different approaches are required, providing a brief justification would strengthen the argument.
- There are several strikethrough lines throughout the document, please remove these for clarity.
Response to comment 2:
We have revised the Introduction section and included a clear definition of chronic brain injury (CBI).
Additionally, new references have been incorporated to strengthen the rationale and support the arguments presented.
(The revised version is attached as a PDF.)
Comment 3 Methods
This section is well written. However, some statements lack proper referencing. For example:
- Lines 338–339
- Lines 347–348
Please ensure that all evidence-based statements are supported with appropriate citations.
Response to comment 3:
|
Thank you for your observation. We have reviewed all statements and now we can assure that citations are more appropriated. |
Comment 4 Discussion
The discussion is well written and clearly presented.
Response to comment 4: We sincerely thank the reviewer for this positive feedback and appreciation of our discussion section. We are glad that it was found to be clear and well presented.
Round 2
Reviewer 1 Report
Comments and Suggestions for Authors
I accept the paper in its current form. The only comment is that the figure numbering is incorrect — the figure labeled as number 5 should actually be number 4.
Author Response
REVIEWER: I accept the paper in its current form. The only comment is that the figure numbering is incorrect — the figure labeled as number 5 should actually be number 4.
ANSWER: Thanks for your comment. We have changed the label of figure 5 to figure 4.